# Toward an Understanding of the Structural and Mechanistic Aspects of Protein-Protein Interactions in 2-Oxoacid Dehydrogenase Complexes

**DOI:** 10.3390/life11050407

**Published:** 2021-04-29

**Authors:** Natalia S. Nemeria, Xu Zhang, Joao Leandro, Jieyu Zhou, Luying Yang, Sander M. Houten, Frank Jordan

**Affiliations:** 1Department of Chemistry, Rutgers, The State University of New Jersey, Newark, NJ 07102, USA; jieyu.zhou@abzena.com (J.Z.); yly19911008@gmail.com (L.Y.); 2Department of Genetics and Genomic Sciences, Icahn School of Medicine at Mount Sinai, New York, NY 10029, USA; joao.leandro@mssm.edu (J.L.); sander.houten@mssm.edu (S.M.H.)

**Keywords:** neurodegeneration, glucose metabolism, enzyme catalysis, protein-protein interaction, hydrogen exchange mass spectrometry, protein cross-linking, protein assembly, molecular modeling

## Abstract

The 2-oxoglutarate dehydrogenase complex (OGDHc) is a key enzyme in the tricarboxylic acid (TCA) cycle and represents one of the major regulators of mitochondrial metabolism through NADH and reactive oxygen species levels. The OGDHc impacts cell metabolic and cell signaling pathways through the coupling of 2-oxoglutarate metabolism to gene transcription related to tumor cell proliferation and aging. *DHTKD1* is a gene encoding 2-oxoadipate dehydrogenase (E1a), which functions in the L-lysine degradation pathway. The potentially damaging variants in *DHTKD1* have been associated to the (neuro) pathogenesis of several diseases. Evidence was obtained for the formation of a hybrid complex between the OGDHc and E1a, suggesting a potential cross talk between the two metabolic pathways and raising fundamental questions about their assembly. Here we reviewed the recent findings and advances in understanding of protein-protein interactions in OGDHc and 2-oxoadipate dehydrogenase complex (OADHc), an understanding that will create a scaffold to help design approaches to mitigate the effects of diseases associated with dysfunction of the TCA cycle or lysine degradation. A combination of biochemical, biophysical and structural approaches such as chemical cross-linking MS and cryo-EM appears particularly promising to provide vital information for the assembly of 2-oxoacid dehydrogenase complexes, their function and regulation.

## 1. Introduction

The human 2-oxoglutarate dehydrogenase complex (OGDHc) is a key enzyme in the tricarboxylic acid (TCA) cycle, which is a common pathway for oxidation of fuel molecules, including carbohydrates, fatty acids, and amino acids. Diminished OGDHc activity and mitochondrial abnormalities have been correlated with numerous neurodegenerative disorders, including Alzheimer’s disease, however, a link between reductions in the mitochondrial TCA cycle enzymes and (neuro) degeneration has not been established so far [1,2,3,4]. Recently, a biallelic pathogenic variant in *OGDH* gene leading to deficient 2-oxoglutarate dehydrogenase (E1o, the first component of the OGDHc, also known as OGDH) was reported in individuals with a neurological disorder resembling mitochondrial disease [5]. While rare in *OGDH*, whole-exome sequencing and rare variant burden analysis determined an overabundance of putative, potentially damaging mutations in the *OGDHL (OGDH-like)* and *DHTKD1* genes across multiple patients with eosinophilic esophagitis (EoE), a chronic allergic disorder that presents in infancy and in adulthood [6]. The *OGDHL* encodes a putative 2-oxoglutarate dehydrogenase-like protein (E1o-like) in the TCA cycle that is tissue-specific and is mainly expressed in brain and liver (Scheme 1) [7]. Furthermore, a homozygous deleterious variant (c.2333C > T; p. Ser778Leu) was recently identified in *OGDHL* and was associated with neuro-degenerative phenotype in patients [8]. The *DHTKD1* gene encodes a less-known homologue of E1o, the enzyme 2-oxoadipate dehydrogenase (E1a, also known as DHTKD1) in the L-lysine degradation pathway (Scheme 1). Genetic studies have linked variants in *DHTKD1* to the (neuro) pathogenesis of several metabolic disorders: α-aminoadipic and α-ketoadipic aciduria (AMOXAD: MIM 204750) [9,10,11], Charcot-Marie-Tooth disease type 2Q (CMT2Q: MIM 615025) [12,13,14] and eosinophilic esophagitis (EoE), a chronic allergic disorder [6].

Pharmacological inhibition of E1a has been proposed as a strategy for substrate reduction therapy to treat glutaric aciduria type 1 (GA1: MIM 231670), a metabolic disorder that is caused by mutations in the *GCDH* gene encoding the mitochondrial protein glutaryl-CoA dehydrogenase (GCDH; EC 1.3.8.6) located downstream of E1a in the L-lysine degradation pathway (Scheme 1) [15].

**Scheme 1 life-11-00407-sch001:**
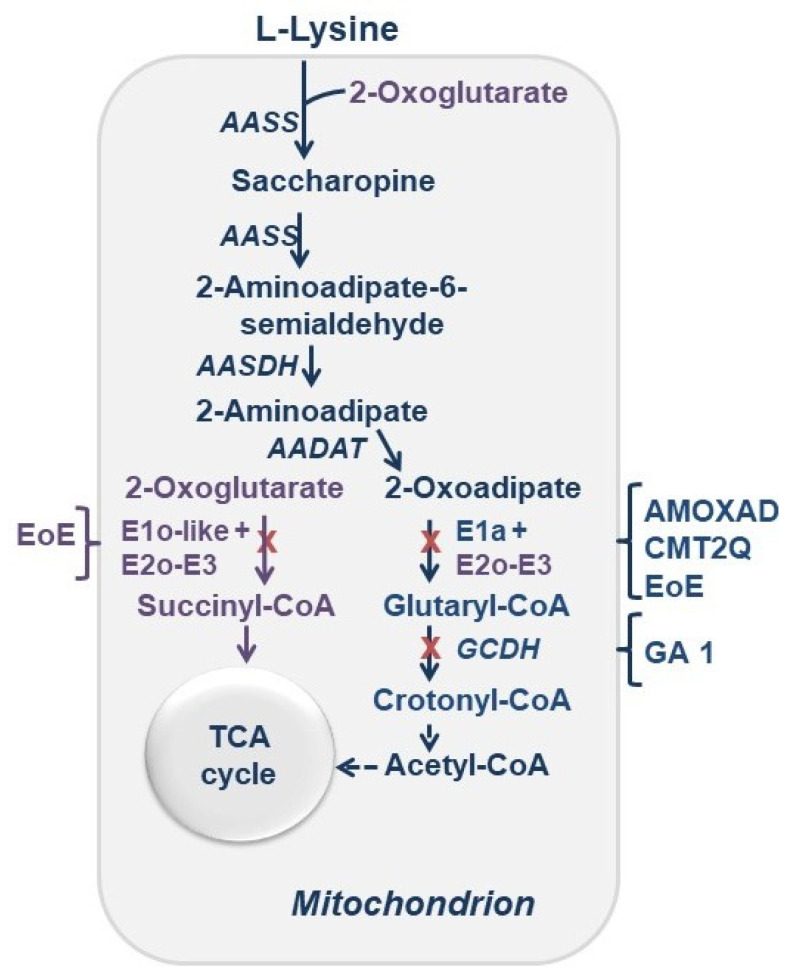
Role of the OGDHc in the TCA cycle and of the E1a in the L-lysine degradation pathway in health and diseases. *AASS*, 2-aminoadipate-6-semialdehyde synthase; *AASDH*, 2-amino-adipate-6-semialdehyde dehydrogenase; *AADAT*, 2-aminoadipate transaminase; *GCDH*, glutaryl-CoA dehydrogenase; E1o-like, E2o, E3-components of the OGDHc; E1a, 2-oxoadipate dehydrogenase; AMOXAD, α-aminoadipic and α-ketoadipic aciduria; CMT2Q, Charcot–Marie-Tooth disease type 2Q; EoE, eosinophilic esophagitis; GA1, glutaric aciduria 1.

Despite the reported genetic findings, no pathophysiologic mechanism has been reported for the disease-associated *OGDH, OGDHL* and *DHTKD1* variants and their impact on E1o, E1o-like and E1a protein structure and function.

Diverse functions of the E1o have been shown to be associated with cancer. As the OGDHc is integrated within mitochondrial functions and represents one of the major regulators of mitochondrial metabolism, it was suggested that it is pivotal in the adaptation of cancer cells to a demanding environment in vivo [16]. Two mitochondrial genes *OGDH* and *LIAS* (encoding lipoic acid synthase) were identified that are involved in the mechanism of regulation in hypoxia-inducible transcription factors (HIFs) under aerobic conditions [17]. Lu et al. suggested that OGDHc mediates SIRT5 (sirtuin 5, an NAD^+^-de-pendent protein deacylase) function, a potential suppressor of cell growth and migration in gastric cancer [18]. The direct interaction between SIRT5 and OGDHc was shown and it was reported that desuccinylation of OGDHc by SIRT5 inhibits OGDHc activity [18]. The authors suggested that SIRT5 suppressed gastric cancer cell growth and migration by inhibiting mitochondrial function and by increasing ROS production via down-regulation of OGDHc activity. Thus, both SIRT5 and OGDHc could be novel therapeutic targets for gastric cancer treatment [18,19]. Several recent studies support the rationale to target individual components of the OGDHc, specifically E1o [20] or E2o [21], or possibly all three components of the OGDHc [22] for cancer treatment [23].

Recent findings suggested that post-translational modifications (PTMs) by succinylation and glutarylation could be one of the mechanisms by which cells adapt to dynamic environmental changes to maintain metabolic homeostasis including regulation of energy production [24,25,26,27,28]. The significance of lysine succinylation and glutarylation has begun to be revealed for nuclear proteins, while its significance for cytosolic and mitochondrial proteins has been examined in only a few proteins [25,29,30]. Thus, glutarylation of the Lys91 in histone H4 in mammalian cells was reported [31]. According to a suggested mechanism, a known histone acetyltransferase KAT2A when coupled with the OADHc, acts as a histone glutaryltransferase in cells, while SIRT7 acts as an “eraser” of the glutarylated Lys91 at histone H4 [31]. On the flip side, when KAT2A is coupled with the OGDHc, it could recognize succinyl-CoA and transfer the succinyl group to the Lys79 of histone H3 [32]. These findings provided evidence for a tight link between metabolism and epigenetic regulation of gene expression by succinylation and glutarylation [31].

Earlier evidence suggested that OGDHc and its E2o component could function as a succinyltransferase for modification of cytosolic and mitochondrial proteins in cultured neurons and in neuronal cell lines and could provide an efficient mechanism to coordinate metabolic pathways at the cellular level [3,27]. Recent findings in vitro suggested that the E2o could also serve as a glutaryltransferase, in addition to functioning as a succinyltransferase, thus providing an efficient mechanism to coordinate the TCA cycle and the L-lysine metabolic pathway at the cellular level in normal state and in disease [33]. Increased lysine glutarylation of mitochondrial proteins in the brain and liver was observed in a GA1 mouse model [25,30]. However, the molecular mechanisms underlying the pathogenesis of GA1 are still poorly understood.

Finally, a novel function of E1o in host cell metabolic adaptation to viral infection was suggested [34]. It was found that in response to viral infection, host cells impair the activity of the *N*^6^-methyladenosine (m^6^A) RNA demethylase leading to increased m^6^A methylation on the E1o mRNA and to reduced mRNA stability. The reduced E1o protein expression led to reduced production of the itaconate intermediate in the TCA cycle, an intermediate which is required for viral replication, thus providing a crosstalk between m^6^A RNA modification and metabolic reprogramming via the E1o-itaconate pathway [34]. The E1o-itaconate metabolic response was suggested as a potential therapeutic target for the control of viral infection [34]. In view of the accumulated data on the physiological importance of the OGDHc and OADHc in human health and disease, and their role as potential therapeutic targets, we first review OGDHc function and discuss a novel alternative mechanism of transthiolacylation catalysis in the active centers of the *E. coli* E2o that would be applicable to all E2 components. Next, we discuss protein-protein interactions identified in the human E1o-E2o, E1o-E3 and E3-E2o binary sub-complexes by hydrogen-deuterium exchange MS (HDX-MS) and chemical cross-linking MS (CL-MS), leading to the remarkable conclusion that the N-terminal region of E1o may constitute a unique dual-subunit-binding domain (DSBD) in human OGDHc, which is recognized by both the E2o and E3 components. Of special interest is recognition of the formation of a hybrid complex between OGDHc and OADHc. In the second half of this review, we discuss the structural insight into the architecture of the human E1a active site during catalysis, of which two independent X-ray structures have been reported recently [35,36]. Taking into consideration both in vitro and in vivo evidence for the interaction between E1a and E2o from two distinct metabolic pathways, we next present the models constructed for the E1o-E2o and E1a-E2o assembly to advance our understanding of protein-protein interactions in human OGDHc and OADHc in the absence of X-ray crystallographic or cryo-EM-based atomic structure of intact OGDHc or OADHc.

## 2. The 2-Oxoglutarate Dehydrogenase Complex

### 2.1. Overview of the Molecular Mechanism of the OGDHc

The canonical human OGDHc is a key enzyme in the TCA cycle, comprising multiple copies of three components, E1o, E2o, and E3. The five principal chemical steps involved in the reactions yielding succinyl-CoA are shown in Figure 1A. The ThDP-dependent E1o catalyzes the decarboxylation of 2-oxoglutarate (OG) (*k*_2_, *k*_3_), followed by reductive succinylation of the E2o (*k*_4_, *k*_5_). The reductive acylation is the important coupling mechanism of the reactions of the E1 and E2 components in 2-oxoacid dehydrogenase complexes, sometimes called substrate channeling. It is generally accepted that lipoyllysyl-E2 (lipoic acid is covalently amidated onto the ε-amino group of a lysine residue of E2 in its N-terminal region, the so called lipoyl domain or LDo) is the oxidizing agent for the enamine product of OG decarboxylation, which is non-covalently assembled with the E1 component [37]. Below we will focus on mechanism for the reductive acylation reaction between the E1 and E2o components and on the synthesis of the acyl-CoA in the active centers of E2o which is a controversial topic.

#### 2.1.1. Mechanism for the Reductive Acylation Reaction

Over the years two mechanisms have been proposed for the reductive acylation reaction: (a) A stepwise mechanism, in which the redox process, i.e., electron transfer, the oxidation of the enamine to the acyl-ThDP with concomitant reduction of the lipoyllysyl-E2 to dihydrolipoyllysyl-E2 is the first step, followed by group transfer [38]; (b) A concerted mechanism, in which the enamine C2α-carbon as a nucleophile attacks the *S*8 atom of the lipoyllysyl-E2, opening the dithiolane ring momentarily and forming a tetrahedral intermediate, thereby transiently linking the two components, the reaction being completed by cleavage of the C2α-C2 bond thus regenerating the ThDP ylide and generating the acyl-*S*8-dihydrolipoate (see Scheme 2 for a concerted mechanism of the OADHc with 2-oxoadipate) [37,39]. According to the second mechanism, electron and group transfer are synchronized, necessitating approach of the lipoyllysyl-E2 to within C-S bond distance to the C2α-carbon of the substrate in the first post-decarboxylation step. Yet, even according to the first mechanism, any plausible transfer of the acyl group to the *S*8-thiolate would require close approach of the lipoyllysyl group to the carbonyl group of the C2-acylThDP. A chemical model developed at Rutgers earlier was more consistent with the second mechanism, proceeding via a tetrahedral intermediate [39]. An appropriate model reaction to determine the rate of reductive acyl transfer from E1 to the lipoyllysyl-E2 has been developed (Figure 1C) [40,41,42,43], which maintains both the chemistry and the inter-component communication due to its specific recognition of E1 [44,45,46]. The full-length E2o is a highly segmented protein, comprising from the N-terminal end, a single lipoyl domain (LDo), linker region and an acyltransferase catalytic domain or E2o core domain (Figure 2A). By using the independently expressed LDo in place of intact E2o, the mass of the acylated and unacylated lipoyl domain could be measured by mass-spectrometry (FT-MS) with high precision, an experiment that at the same time enabled calculation of the rate constant (*k*_4_ in Figure 1A) for acyl transfer between the E1 and E2 components [43].

**Figure 1 life-11-00407-f001:**
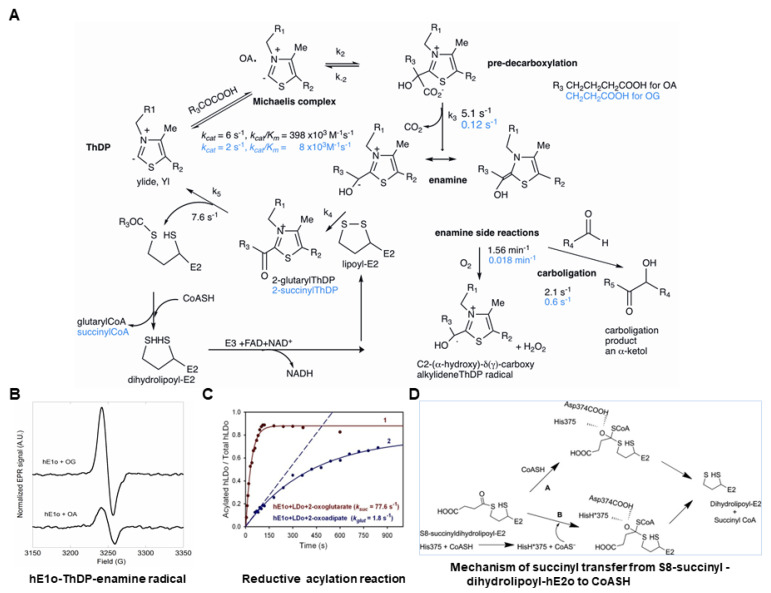
(**A**) Mechanism of the OGDHc with 2-oxoglutarate (in *black*) and 2-oxoadipate (in *blue*) as substrates. (**B**) X-band EPR spectra of the ThDP-enamine radical species generated on E1o OG and OA. The radical species generated on E1o from OA is ~3 times lower in concentration compared to that from OG (0.9 μM with ~0.2% occupancy of E1o active centers). (**C**) Time dependence of reductive acylation of the lipoyllysyl groups on LDo by E1o from OG or OA could be analyzed by FT-MS. The fraction of reductively acylated LDo versus total LDo (acylated plus un-acylated LDo) when plotted versus time, allowed evaluation of the rate of acyl transfer from E1o to E2o (*k_4_* in (**A**)). (**D**) Mechanism of succinyl transfer from *S*8-succinyldihydrolipoyllysyl-E2o to CoASH with suggested role of His375 and Asp374. Pathway A: direct attack by the conjugate base thiolate anion of CoASH assuming a low pK_a_ for the CoASH, or by the thiol form itself. Pathway B: initial proton transfer from CoASH to His375 forming the conjugate base CoAS^-^, which is the attacking agent. Both pathways then proceed by an oxyanionic tetrahedral intermediate [41].

#### 2.1.2. Synthesis of Succinyl-CoA

The reductive succinylation reaction is followed by *trans*-thiolesterification of the succinyl group onto CoA in the active centers of hE2o (i.e., in the catalytic domain, CDo), generating succinyl-CoA (Figure 1A). The enzymatic mechanism responsible for synthesis of succinyl-CoA has been elucidated for the *E. coli* E2o (Figure 1D) [41], a mechanism that is applicable to all E2o components due to highly conserved structure of the reported E2o active centers [47,48,49,50,51,52]. There are two likely mechanisms to account for the transfer of an acyl group between two thiols in Figure 1D. A general acid-base mechanism would suggest that the catalytically important His375 of the *E. coli* E2o catalytic domain converts the thiol group of the attacking nucleophile (CoASH) to a thiolate anion (CoAS^−^) as depicted on pathway B in Figure 1D. The activated thiolate then attacks the carbonyl atom of the succinyldihydrolipoyl-E2o to form a tetrahedral intermediate, which is stabilized by the hydroxyl side chain of Thr323 [48]. This mechanism was suggested based on analogy with a mechanism developed for chloramphenicol acetyltransferase [53,54,55]. However, the catalytic efficiency (*k_cat_*/*K_m_*) of the His375Ala substituted *E. coli* E2o was reduced by only 54-fold compared to unsubstituted *E. coli* E2o, while a 9 × 10^5^-fold reduction in catalytic efficiency was determined for the analogous His195Ala substitution in chloramphenicol acetyltransferase, suggesting different functions for the highly conserved histidine residues on the two enzymes [41]. As an alternative to acid-base catalysis, a direct attack by a thiolate anion on the thiol ester carbon had been suggested, forming a symmetrical tetrahedral oxyanionic intermediate, in which the central carbon atom is flanked by two C-S bonds with nearly equal probability for cleavage [41]. Evidence was provided that His375 and Asp374 play a role in the stabilization of this symmetrical tetrahedral oxyanionic intermediate by formation of two hydrogen bonds, rather than in acid-base catalysis (pathway A in Figure 1D). An important conclusion from these studies is that succinyl transfer to CoA and release of succinyl-CoA, rather than the reductive succinylation reaction, is the rate-limiting step [41]. Next, the E3 re-oxidizes dihydrolipoyllysyl E2o with concomitant reduction of NAD^+^ to NADH (Figure 1A). The E3 is shared by all members of the 2-oxo acid dehydrogenase complex family in mammalian cells.

The human OGDHc is also recognized as a significant source of superoxide radical anion and H_2_O_2_ (reactive oxygen species, ROS) that could lead to oxidative stress in mitochondria [56,57,58,59,60,61,62,63,64]. While earlier this function was assigned to the E3 component in the reverse direction [56,57,59,61], formation of the ThDP-enamine radical species in the active centers of E1o from 2-oxoglutarate and 2-oxoadipate in the physiological direction was demonstrated by electron paramagnetic resonance spectroscopy and represents an “off-pathway” side reaction comprising less than 1% of “on-pathway” reactivity (Figure 1B) [43,60,63,64], as suggested earlier for the *E. coli* E1o [65].

**Scheme 2 life-11-00407-sch002:**
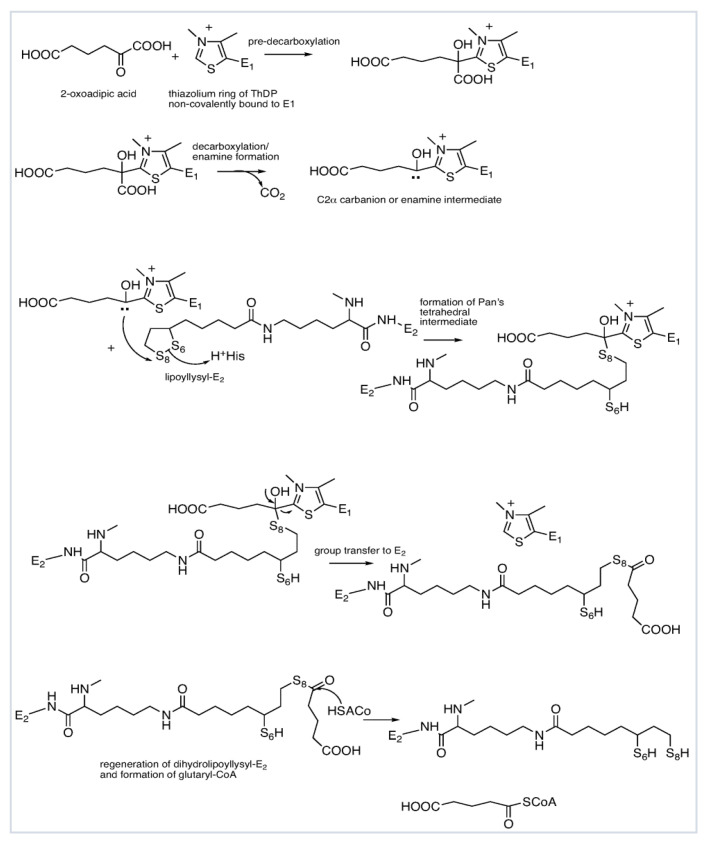
Mechanism of the OADHc (comprising multiple copies of the E1a, E2o and E3 components) showing the formation of the enamine intermediate on E1a from 2-oxoadipate and reductive glutarylation reaction between E1o and E2o according to a concerted mechanism [39].

### 2.2. Protein-Protein Interactions in OGDHc

#### 2.2.1. Interactions in Binary E1o-E2o, E1o-E3 and E2o-E3 Subcomplexes

A typical approach to define protein-protein interactions in multi-enzyme complexes, is to first determine the structure of the individual components. Currently, there is no X-ray structure of the human E1o. There are, however, X-ray structures available for the N-terminally truncated *E. coli* E1o_Δ77_ (39% identity, missing the N-terminal 77 amino acids) [66], the N-terminally truncated *Mycobacterium smegmatis* α-ketoglutarate decarbo-xylase (*Ms*KGD_Δ115_, 41% identities, missing the N-terminal 115 amino acids) [67], the *Ms*KGD_Δ360_ catalytic domain, which showed overall structural similarity to the *E. coli* E1o [68], and the *Ms*KGD_Δ360_ catalytic domain in complex with phosphonate analogues of OG [69]. X-ray structures were recently reported for the N-terminally truncated human E1a (residues 45-919, missing the N-terminal 44 residues) at 1.9 Å [35] and for human E1a (residues 25-919, missing the N-terminal 24 residues) at 2.25 Å resolution (38.5% identities between E1o and E1a) [36]. Single-particle cryo-EM reconstruction of the E2o inner core domain at 4.7 Å global resolution [35] has been reported showing 24 E2o catalytic domains assembled as eight trimers and positioned at each of the eight vertices of the cubic core with octahedral symmetry (Figure 2B). However, there is no atomic resolution structure reported for any of the 2-oxo acid dehydrogenase complexes. In the absence of the direct insight into assembly of the components into OGDHc, a multipronged approach has been employed including fluorescence, HDX-MS and CL-MS studies which allowed evaluation of the strength and loci of interaction in binary E1o-E2o, E1o-E3 and E2o-E3 sub-complexes [70]. Among the remarkable findings is that in the binary sub-complexes, strong interactions (*K_d_* in the 0.04–0.14 μM range) were demonstrated between the E1o and E2o components, but not between the E2o and E3. Importantly, two peptides from the N-terminal region of E1o comprising residues ^18^YVEEM^22^ and ^27^LENPKSVHKSWDIF^40^ were identified that represent the most likely candidates for the interaction of E1o with both E2o and E3 [70]. The important role of the E2o region comprising residues from both the E2o core domain and the linker region was identified for the first time as critical for E1o-E2o interactions (Figure 2C,D) and represents a unique E2o-binding mode in human OGDHc [70]. In contrast, there was no evidence for E2o-E3 interaction indicating that there was only a transient interaction between these two components, too weak to be captured by the methods applied. These findings are in accord with the accepted OGDHc mechanism in Figure 1A, where with each turnover, the reduced lipoyl domain (dihydrolipoyllysilyl-group on E2o) must be re-oxidized by the FAD on E3 [71] In summary, the N-terminal region of E1o (residues 18–40) could constitute a unique dual-subunit-binding domain (DSBD) in human OGDHc, which is recognized by both the E2o and E3 components, suggesting that an initial formation of the uniquely strong E1o-E2o interaction could facilitate assembly with E3 into OGDHc, a hypothesis that needs to be confirmed in further studies.

#### 2.2.2. The Assembly of E3 into OGDHc

Related to the assembly of E3 into OGDHc, a novel structural component of the mito-chondrial OGDHc named as Kgd4 (Ymr31) was identified in yeast, previously described as a part of the mitochondrial ribosome with a role in recruiting the E3 subunits to OGDHc [72]. A Kgd4 homologue was also identified as part of the murine OGDHc (*MRPS36*) [72]. A model was suggested for organization of the mitochondrial OGDHc where Kgd4 binds to the preformed E1-E2 core to recruit the E3 subunit into the complex while in the absence of Kgd4 the binding of E3 to the core is dramatically reduced [72]. However, in recently reported studies by Houten’s group, the transient transfection of *MRPS36* encoded protein together with the E1o (E1a), E2o and E3, did not increase the activity of the OGDHc or of OADHc in HEK-293 cell lysates with OG and OA as substrates [73], indicating a different assembly of the E3 in these human complexes. Notably, the human E3 which is shared by all members of the 2-oxoacid dehydrogenase complex family in mammalian cells has a different binding mode in the human pyruvate dehydrogenase complex (PDHc), where the pyruvate dehydrogenase component (E1p) binds to the E1p-binding domain on E2p (peripheral subunit-binding domain, PSBD), while the E3 binds to the E3-binding protein (E3BP, formerly known as Protein X), a component with no known catalytic function [74]. The E2p and E3BP (typically co-expressed) form the inner core of the mammalian PDHc according to two distinct “substitution” models proposed: 48 subunits of E2p + 12 subunits of E3BP or 40 subunits of E2p + 20 subunits of E3BP in the 60-meric inner core [75,76,77,78,79,80]. Structure determination of the human E2p inner core at 3.1 Å atomic resolution [79], and of the full-length human E2p-E3BP assembly at 6.3 Å resolution [80] have been reported recently by using cryo-EM and single particle reconstruction (see below). More questions have been raised and need to be answered regarding the E1o-E2o interaction in human OGDHc since the recent finding that the human 2-oxoadipate dehydrogenase (E1a) of the L-lysine, L-hydroxylysine and L-tryptophan catabolic pathway and the E1o of the OGDHc in the TCA cycle share the same E2o component and can utilize the recycling universally used E3 component for their function, while earlier the formation of hybrid complexes between OGDHc and PDHc had been reported.

**Figure 2 life-11-00407-f002:**
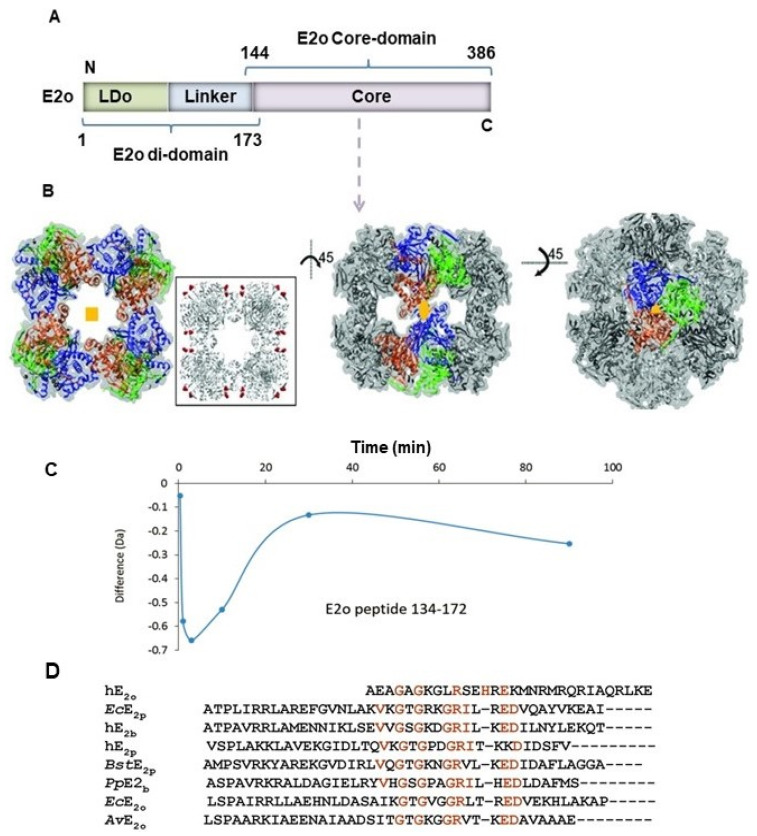
(**A**) Domain structure of the human E2o showing the lipoyl domain (LDo) and the E2o core or catalytic domain (CDo) connected by a linker region. (**B**) Electron micrograph reconstruction of the human E2o core domain structure at 4.7 Å resolution [35]. Figure reproduced with permission of the International Union of Crystallography). The trimer building blocks (the individual subunits are shown in *blue*, *green* and *orange*) are assembled into the 24-meric core via four-fold (left), two-fold (middle) and three-fold symmetry axes, respectively. Insert shows the first residue 219 (red spheres) from each subunit of the 24-mer E2o core which is exposed to the surface of the E2o core. (**C**) On human E2o, the region comprising residues from the core domain (in bold) and E2o linker region (^144^AEAGAGKGLRSEH**REKMNRMRQRIAQRLKE**^174^) displayed a significant decrease in the level of the deuterium uptake during the first 3 min upon E1o binding, suggesting a unique subunit-binding mode in human OGDHc assembly, where the E2o core domain also participates in the interaction with E1o. (**D**) Sequence alignment of the human E2o linker-core region involved in interaction with E1o with some known E1-binding domains in E2 components identified conserved residues, indicating that E2o shares some but not all sequence features of other subunit-binding domains involved in E1o binding. The abbreviations are denoted: human (h), *Escherichia coli* (*Ec*), *Bacillus stearothermophilus* (*Bst*), *Pseudomonas putida* (*Pp*), and *Azotobacter vinelandii* (*Av*), 2-oxoglutarate dehydrogenase complex (o), pyruvate dehydrogenase complex (p), and branched chain 2-oxoacid dehydrogenase complex (b). Alignment of multiple sequences was carried out using the Clustal Omega program with default settings [70].

### 2.3. The Newly Revealed Feature of the OGDHc in the TCA Cycle, the Propensity to Form a Hybrid Complex with E1a Ftom the L-Lysine Degradation Pathway

The first evidence for the formation of a hybrid 2-oxo acid dehydrogenase complex involving the E1o, E2o and E3 components of the OGDHc and the E1p component of the PDHc had been reported more than three decades ago for *E. coli* cells, with no physio-logical importance attributed to such a hybrid complex [81]. Later, the unique properties with respect to their protein components and three-dimensional organization of the PDHc and OGDHc from Gram-positive bacteria such as *Mycobacterium tuberculosis* (*Mtb*) and *Corynebacterium glutamicum* were reported [82,83]. There is an E2p component in these bacteria, which on assembly with E1p and E3 converts pyruvate to acetyl-CoA in the overall PDHc reaction, while there has been no such E2o component identified for OGDHc. Instead, the OdhA from *Corynebacterium glutamicum* and *Mtb*Kgd assemble as a single fusion protein with two major domains, the N-terminal acyltransferase-like domain and the E1 domain [83]. It was demonstrated that OdhA from *Corynebacterium glutamicum* catalyzes both the E1 and E2 reactions and could convert 2-oxoglutarate to succinyl-CoA. However, it is totally dependent on the E2p component of the PDHc as a source of its lipoyl groups. These findings suggested that E1p and OdhA shared the E2p component, in addition to sharing E3, and could form a hybrid complex consisting of E1p, E2p, E3, along with OdhA, in accord with earlier evidence from co-purification experiments [82]. Later, the presence of a hybrid OGDHc/PDHc was also suggested for the *Mycobacterium smegmatis* α-ketoglutarate decarboxylase (*Ms*KGD, E1o) which on reconstitution with dihydrolipoamide acyltransferase (DlaT, the Rv2215 gene product) and E3 revealed OGDHc activity according to formation of succinyl-CoA from OG and CoA [67]. It was proposed that formation of a hybrid OGDHc/PDHc assembly is relevant to stimulation of the *Ms*KGD activity by acetyl-CoA, which is produced by PDHc, and may be important for coordination of the OGDHc/PDHc activities [67,84].

Recently evidence was obtained for a unique property of the human E1a in the L-lysine degradation pathway, where in the absence of a cognate dihydrolipoamide glutaryltransferase component (E2a, no candidate gene has been identified to date), the E1a recruits the E2o and E3 components of the OGDHc for its function in vitro [42,43] and forms a hybrid 2-oxo acid dehydrogenase complex in vivo containing the E1o, E2o and E3 components along with E1a [73]. There is also experimental evidence of a direct interaction between the E1a and E2o components: (i) On co-expression of human E1a and E2o in *E. coli* and in insect Sf9 cells, the E1a_45-919_ was co-purified with the E2o_68-453_ protein, signaling formation of a high-affinity sub-complex [35]. (ii) Indeed, on titration of the fluorescent N-(1-pyrene) maleimide labeled truncated human E2o (E2o^1-173^ di-domain and E2o^144-386^ inner core domain) by E1a, dissociation constants in the 1–4 μM range were determined, comparable to that for E1o-E2o sub-complex indicating a strong interaction between the E1a and E2o proteins [35,85]. (iii) It was further demonstrated that human E1a was co-immunoprecipitated from DBA/2J mouse liver and from control HEK-293 cells with the human E2o, E3 and E1o component as well, supporting the existence of a hybrid 2-oxo acid dehydrogenase complex [73]. The crosstalk between the OGDHc in the TCA cycle and OADHc in L-lysine catabolism is of potential functional relevance since OGDHc can serve as an alternative source of glutaryl-CoA formation in GA1 when E1a function is abolished, which limits the use of E1a-specific inhibitors for substrate-reduction therapy [15,73]. It could furthermore offer a likely explanation for the rather mild or absent phenotype in patients with AMOXAD due to *DHTKD1* mutations [9,10,11]. In addition, the crosstalk may provide a mechanism that could compensate for defects in mitochondrial energy metabolism due to the *DHTKD1*^Y486*^ mutation in CMT2Q [86] and explain an association between *DHTKD1* and *OGDHL* encoded variants and mitochondrial dysfunction related to EoE [6].

## 3. Structural Insight into the E1a Active Center

The idea that a specific lipoyl domain of E2 is recognized by its cognate E1, was generally accepted for a long time. A major recent advance in the field has been the realization that the E1a in the L-lysine degradation pathway could share the same E2o and E3 components with the E1o from the TCA cycle for their function in two distinct metabolic pathways [42,43,64]. Next, it was shown that the E1a has about 40-fold preference in catalytic efficiency with OA versus OG [43], suggesting that the E1a active center has evolved to accommodate the slightly longer OA compared to OG (one additional CH_2_ group) substrate. These are important determinants for the rational design of E1a inhibitors that could be useful for diseases such as GA1 [36].

Below we discuss the structural insight into the architecture of the human E1a active site during catalysis, as it is a less studied enzyme in the family of 2-oxoacid dehydro-genase complexes. The thiamin diphosphate (ThDP) dependent E1a is the first component of the OADHc [7,9,10,11,12,13], the totality of which carries out the reactions forming glutaryl-CoA according to the overall reaction in Equation (1) and with detailed chemistry in Equations (2)–(5) and Scheme 1 and Scheme 2:2-oxoadipate +NAD^+^ + CoA → glutaryl-CoA + NADH + H^+^+ CO_2_(1)
2-oxoadipate + E1a → C2-(α-hydroxy)-γ-carboxybutylidene-ThDP-E1a (the enamine intermediate) + CO_2_(2)
(C2-(α-hydroxy)-γ-carboxybutylidene-ThDP-E1a + lipoyl-E2o → *S*8-glutaryl-dihydrolipoyl-E2o (reductive glutarylation)(3)
*S*8-glutaryldihydrolipoyl-E2o + CoA → glutaryl-CoA + dihydrolipoyl-E2o(4)
dihydrolipoyl-E2o + E3 + NAD^+^ → lipoyl-E2o + NADH(5)

The recently solved X-ray structures of the human E1a at 1.9 Å (PDB code 6sy1) [35] and at 2.25 Å resolution (PDB code 6U3J) [36] revealed its structural homology to *Ms*KGD (*Ms*KGD_Δ115_; PDB code 2XT6; 38% identity) and to *E. coli* E1o (PDB code 2jgd; 38% sequence identity) [66]. The E1a forms a tight homodimer of over 200 kDa, common to all ThDP-dependent enzymes, burying a more than 5000 Å^2^ area of monomeric accessible surface at the dimer interface (Figure 3A) [36]. Each E1a subunit is structurally composed of four distinct domains displaying the common fold of ThDP-dependent enzymes (Figure 3, top) [35]. The ThDP cofactor is bound between the two subunits and is supported by highly conserved hydrogen bonds and hydrophobic interactions, including Asp333 and Asn366 from the known ThDP-binding fold (Figure 3B) [87,88]. From the reported structural studies several amino acid residues in the E1a active center have been suggested to interact with OA, residues which are not conserved with those involved in OG binding in E1o’s: more polar residues including Tyr190 (Phe506 in *Ms*KGD; Phe227 in *E. coli* E1o), Tyr370 (Phe682 in *Ms*KGD; Phe394 in *E. coli* E1o), a less bulky Ser263 (Tyr578 in *Ms*KGD; Tyr297 in *E. coli* E1o), and a conserved Asp707 (Asp1019 in *Ms*KGD; Asp688 in *E. coli* E1o) [35,36]. To gain further structural insight into the E1a active center, models have been built of its active center with covalent reaction intermediates derived from OA based on the reported X-ray structures of human E1a [35,36] and *Ms*KGD [68], as well as the mechanism of OGDHc with OA presented in Scheme 2. According to the model studies, the pre-decarboxylation covalent adduct at the thiazolium C2 position of ThDP is stabilized by the salt bridge between its leaving carboxylate group and the imidazole group of His223 (His539 in *Ms*KGD) and by a hydrogen-bonding interaction between the 2-oxo group of OA and the imidazole group of His708 (His1020 in *Ms*KGD) (Figure 3C). The ε-carboxyl group of the OA in the pre-decarboxylation intermediate is sandwiched between Tyr190 and Tyr370 to form polar interactions in accord with structural findings (Figure 3C) [35]. Interestingly, the Tyr190Phe E1a substitution reduced the catalytic efficiency of the human E1a measured in the model reaction with OA by ~5.7-fold with no activity detected with OG, indicating that Tyr190 is not crucial for OA binding [36]. In comparison, the ThDP-enamine covalent post-decarboxylation intermediate in Figure 3D engaged new hydrogen-bonding interactions where the substrate ε-carboxyl group now interacts with imidazole groups of His264 and His708 and a side chain hydroxyl group of Ser288, suggesting possible rearrangement of the active site environment (Figure 3D). Earlier, two distinct conformations of the post-decarboxylation intermediate were identified by X-ray crystallography for the *Ms*KGD [68], however, no such information was available for other homologues of E1o so far. Next, on modeling of the E1a active site with the lipoyllysyl-arm of E2o (see Figure 3E for reductive acyl transfer mechanism and Figure 3F), the residues His708 and His 435 are sufficiently proximal to the incoming lipoyllysyl-arm of E2o to act as a general acid catalyst during the reductive glutarylation of E2o, as suggested by the Pan-Jordan model in Scheme 2 [39]. This is also in agreement with findings in *P. putida* E1b [89], where His312α and H131β would serve as potential proton donors to the two sulfur atoms of the lipoyllysyl group of E2 during the reductive acyl transfer step. Through amino acid sequence alignment, the *P. putida* E1b His312α and H131β residues align well with His435 and His708 in human E1a. An understanding of the possible interactions in the E1o-E2o and E1a-E2o binary sub-complexes at the atomic level remains challenging so far.

## 4. Similarities and Differences between the E1o-E2o and E1a-E2o Assembly According to Mass Spectrometric Studies

To identify the similarities and differences between the E1o-E2o and E1a-E2o sub-complex assembly, protein-protein interactions in the binary E1a-E2o sub-complex were analyzed by HDX-MS. Importantly, the N-terminal region of E1a comprising residues 24–47 experienced significant reduction in deuterium uptake in the E1a-E2o sub-complex (Figure 4A) [85] similar to that observed with the E1o-E2o sub-complex [70]. Also, the region surrounding the ThDP- and Mg^2+^-binding site, including peptides 485–505, 506–518 and 646–664 in E1a displayed a lower relative deuterium incorporation, suggesting less accessibility to the deuterium on interaction with E2o. However, the peptide comprising residues 407–415 in the same region of E1o displayed modestly increased deuterium uptake [70,85]. In Figure 4B, peptides that become less open for deuterium uptake on interaction with E2o are color-coded in *red* on the modeled E1 structure. More differences were detected in the interaction of E1a with E2o in the E1a C-terminal region, where peptide 847–874 was protected from deuterium uptake [85], while peptide comprising residues 865–887 in E1o was exposed to deuterium exchange and displayed a significant increase (>1 Da) in deuterium uptake [70]. These HDX-MS studies led to a major conclusion that, the N-terminal region of both E1o and E1a is a candidate for binding with E2o.

To further elucidate potential loci of interaction in the binary E1a-E2o sub-complex, chemical cross-linking MS (CL-MS) experiments were carried out by using two cross-linkers: 1,1′-carbonyldiimidazole, a zero-length cross-linker (CDI, a spacer length of 2.6 Å and Cα-Cα distance of ~16 Å to be bridged) and disuccinimidyl dibutyric urea (DSBU, Cα-Cα distance of ~27 Å to be bridged). The majority of cross-linked residues identified in E1a are located in the N-terminal region (Lys^37^, Lys^148^ and Lys^188^), in the ThDP-Mg^2+^-binding region (Lys^300^ and Lys^450^), and in the C-terminal region (Lys^818^, Lys^826/827^, and Lys^852/854^) (Figure 5). Lysine residues from those regions were mostly cross-linked to E2o lysine residues from the lipoyl domain (Lys^24^, Lys^43^, Lys^66^, Lys^78^, and Lys^85^), from the linker region (Lys^150^, Lys^159^), and from the inner core domain (Lys^240^, Lys^286^, Lys^342^, Lys^371^) (Figure 5 lower panel). It was notable that a great number of interactions between the C-terminal region of E1a and the lipoyl domain of the E2o have been formed suggesting that the C-terminal region of E1a may play an important role in E1a-E2o interactions, particularly, in the glutaryl transfer from the E1a active center to the E2o. This conclusion was confirmed by studies with the G729R E1a variant identified in AMOXAD disease [85]. Thus, the HDX-MS and CL-MS findings provided several layers of information on E1o-E2o and E1a-E2o assembly in Figure 5. First, the studies strongly support the role of the N-terminal region of the human E1o and E1a to interact with E2o, which has now become a general binding mode for E1′s with α_2_ quaternary structure [90,91,92,93]. Second, in the E2o component, the role of the E2o-core domain in interaction with the E1′s was elucidated for the first time. Earlier, this function was assigned mainly to the E1 subunit-binding domain. Third, the role of the C-terminal region of E1a in substrate channeling between the E1a and E2o was identified.

## 5. Structural Modeling of E1a-E2o and E1o-E2o Interactions Using Distance Constraints from

To gain a deeper understanding of the human E1o-E2o and E1a-E2o interactions in binary sub-complexes, cross-linked guided modeling was performed by using web servers and structural prediction algorithms reported in the literature [94,95,96,97,98,99,100,101,102,103]. In the first stage, preliminary models were generated for the human E1o, E1a and E2o by using the I-TASSER algorithm (see Figure 6A for workflow diagram). The E1a structure was built by using the recently reported X-ray structure of the E1a_25-919_ [36]. The E1o structure was modeled by using the reported X-ray structure of the *Ms*KGD (PDB: 2XT6; 41% identity) as the search template. The E2o structure was built by using multiple templates with I-TASSER (PDB: 1SCZ; 6H05; 3MAE). In the second stage, the incompatible intra-protein (or intra-component) cross-links were filtered by using the Xlink Analyzer [98] in visualization system UCSF Chimera (version 1.11) [99] by applying the distance threshold of 35Ǻ for DSBU and of 25 Ǻ for CDI. The Euclidean cut off was calculated as the sum of the length of the two extended lysine chains (2 × 5.5 Å) plus the spacer length (2.6 Å and 12.5 Å for CDI and DSBU, respectively) with an additional 7.6–12.6 Å allowed for the conformational dynamics [100]. The compatible intra-component cross-links were utilized as input with an initial model for modeling refinement, which is especially important for proteins with unknown structure, such as human E1o. The utilization of two different cross-linkers can facilitate these refinement steps. The best models of E1o, E1a and E3 were selected according to the confidence score (C-score) and by how well the intra-component cross-links fit to a model [95]. In the third stage, the resulting protein models were used as input for protein-protein docking by using the HDOCK server [101] and the docking was assisted by the inter-component cross-links identified for the E1a-E2o and E1o-E2o sub-complexes by CL-MS (Figure 5; Figure 6B,C). Also, during the docking stage, the DSBU cross-linked residues were mostly employed as distance restraints since they could provide a broader distance range for protein dynamics compared to the CDI cross-linker (see insert in Figure 6B,C). When the same two Lys residues were found to be cross-linked by both CDI and DSBU, the CDI’s distance restraints were employed. The best model was selected from the top 10 solutions by taking into consideration the previous HDX-MS findings and keeping in mind the current understanding of the catalytic mechanism of the OGDHc and E1a. First, according to HDX-MS studies, the N-terminal region of E1o (residues 27–40) and of E1a (residues 24–47) displayed significant deuterium uptake decrease upon interaction with E2o, suggesting that the N-terminal region of both proteins may be involved in interaction with E2o. Second, the lipoyl domain of the E2o should be near the E1o/E1a active center at a position such that it can transfer the succinyl/glutaryl group from the E1o/E1a active center to the E2o, thus providing the substrate channeling pathway in accord with the catalytic mechanism. It needs to be noted that the resulting models for the E1o-E2o and E1a-E2o sub-complex assembly in Figure 7 share some similarities. 

Thus, in the E2o protein the inner core region comprising residues 227–453 binds on the ‘front side’ of the E1o and E1a proteins providing protection for the N-terminal region of the E1o (residues 27–40) and E1a (residues 24–47) from H/D exchange in accord with HDX-MS findings where the N-terminal region of E1o and E1a displayed significant deuterium uptake decrease on interaction with E2o. Also, the lipoyllysyl-arm of the E2o comprising residues 68–128 of the lipoyl domain and residues 129–226 of the linker region preceding the catalytic domain, is swinging around the E1o/E1a ThDP-Mg^2+^-binding site to provide the catalytic mechanism for succinyl/glutaryl transfer between the E1o/E1a and E2 components. On the other hand, three regions of the E2o core domain with α-helix secondary structure comprising residues 191–208 from the linker region and residues 370–386 from the E2o core are located proximal to E1o, while the region comprising residues 273–289 from the E2o core domain points away from the E1o (Figure 7B, Left). In the E1a-E2o model, a “clockwise twist” shifts the α-helix from the linker region (residues 191–208) away from E1a and brings the α-helix from the E2o core domain (residues 273–289) closer to E1a, while the α-helix from the E2o core (residues 370–386) approaches the N-terminal region of E1a (Figure 7B, Right). These findings help us to understand how E2o could differentiate between the E1o and E1a and suggest further even more challenging studies to obtain an atomic structural model for the OGDHc and its hybrid complex assembly with E1a.

## 6. Application of Cryo-EM to Gain Insight into the Architecture of the 2-Oxo Acid Dehydrogenase Complexes

A thorough identification of protein-protein interactions in the family of the 2-oxo acid dehydrogenase complexes remains a challenge. Early studies of the architecture of the 2-oxo acid dehydrogenase complexes by analysis of electron cryo-microscopy images of the *E. coli* pyruvate dehydrogenase complex (PDHc) and OGDHc were reported three decades ago [105]. The important observation from these studies was that the E2 components in both complexes exist as 24-mer cubic assemblies that form the structural cores of the complexes. Multiple copies (12–24 subunits) of the E1 and E3 bind to the surface of the E2 core and are separated from the core surface by a gap of ~3–5 nm [105]. Some 15 years later, the molecular organization of the *E. coli* PDHc and OGDHc was reinvestigated by using cryo-electron tomography and it was demonstrated that the E1 and E3 components in these complexes are flexibly tethered ~11 nm away from the E2 core with no significant variations in the E2 core dimensions [106]. Recently, a 3D reconstruction of the human E2o core at 4.7 Å resolution was generated by single-particle cryo-EM [35]. The EM reconstruction showed 24 C-terminal core domains grouped as eight trimers and positioned at each of the eight vertices of the cubic cage with octahedral symmetry [35]. There was no electron density found for the other parts of the E2o, including the N-terminal lipoyl domain and much of the linker region, suggesting that those regions are highly flexible and dynamic [35]. The human E2o core structure generated was similar those reported earlier for the *E. coli* E2o core [47], the *E. coli* E2p core [52] and for the *A. vinelandii* E2p core [49], all obtained by X-ray crystallography. In the 3D reconstruction of the co-expressed human E1a-E2o sub-complexes by single-particle cryo-EM, there was no density evident for the E1a protein, indicating that their interaction could be short lived [35]. A molecular weight of the human E1a_45-919_-E2o_68-453_ sub-complex of 2.45 MDa was derived from small-angle X-ray scattering (SAXS) experiments, suggesting a stoichiometry of one E1a dimer per one E2o trimer for their binding [35], similar to that previously deduced for the human E1o-E2o sub-complex [70].

In contrast to 2-oxoglutarate dehydrogenase complexes, cryo-EM studies of the pyruvate dehydrogenase complexes (PDHc’s) from Gram-positive bacteria such as *Bacillus stearothermophilus* [107], yeast [108,109,110], bovine PDHc [111,112], and human PDHc [51,79,80] revealed a 60-mer E2p core with the morphology of a pentagonal dodecahedron and icosahedral symmetry. The first reconstruction of a three-dimensional structure of E2p was reported for the icosahedral truncated E2p (*t*E2p) core from *Saccharomyces cerevisiae* (*S. cerevisiae*) which lacks the lipoyl domain and the E1-binding domain [108,109]. The cryo-EM structure of the *S. cerevisiae t*E2p core generated at 25 Ǻ resolution revealed that the E2p catalytic domains are arranged in cone-shaped trimers at each of the 20 vertices of the dodecahedron structure which are interconnected by 30 bridges [108,109]. The cryo-EM studies also revealed flexibility (size variability) in the arrangement of the *S. cerevisiae* E2p core that is due to conformational changes in the trimers and in the bridges that connect adjacent trimers suggesting a flexible “breathing” core, which may be important for multienzyme complex function, such as channeling of intermediates between the active centers of the individual components [112]. Next, a 3-D reconstruction of the *S. cerevisiae* E2p core in sub-complex with E1p’s generated by cryo-EM at ~30 Ǻ resolution suggested that the length of the E2p inner linkers that anchors the E1p tetramers from the outer shell to the E2p core also may vary from ~50–75 Ǻ depending on the occupancy of the outer shell by E1p’s, contributing to the overall dynamics of the *S. cereviciae* PDHc [110]. The ~70 Å resolution structure generated for the E1p outer shell was significantly lower compared to the E2p core and was less reliable for interpretation of the E1p’s arrangement around the E2p core [110]. The high resolution cryo-EM and single-particle reconstruction enabled definition of 3D structures of the human full-length E2p at 15 Ǻ resolution and of the *t*E2p core (devoid of lipoyl domains and E1p-binding domains) at 8.8 Ǻ and 3.1 Ǻ resolution that also enabled visualization of the secondary structure elements within the *t*E2p core, including the α helices and β sheets [51,79]. The reconstructed structures were similar and consisted of 60 inner core domains arranged as 20 trimers into pentagonal dodecahedron. However, the E2p linker region that connects E1p’s to the E2p core domain was invisible due to its high flexibility [79]. The human E2p core similar to that for *S. cerevisiae* E2p core, exhibited variation in particle size suggesting its flexibility [51].

In eukaryotes, the complexity of the E2p core is increased, being formed by the E2p and E3-binding protein (E3BP) components, which assemble into a 60-mer central core with icosahedral symmetry [113]. Two alternative models have been suggested for assembly of E2/E3BP: 48 copies of E2 and 12 copies of E3BP or 40 copies of E2 and 20 copies of E3BP, where the E3BP substitutes for E2 within trimers [79]. The homology model of the E3BP inner core supports the structure of the preformed heterotrimer with one copy of E3BP and two copies of E2 [79]. Recently, the structure of the recombinant human E2-E3BP core was determined by cryo-EM at 6.3 Å resolution [80]. The final 3D map showed the E2-E3BP inner core where the E2 and E3BP subunits remained undistinguished. The N-terminal lipoyl domains, E1 binding domain and linker regions were invisible due to their high conformational flexibility. The negative stain electron microscopy, native and cross-linking MS gave evidence of the possible structural asymmetry in distribution of the E1p and E3p components at the periphery of the E2p core that suggested a “division-of-labor”mechanism that could be modulated by the presence of substrate [80]. Currently, it is evident that even high-resolution structures of human E2p core and E2p-E3BP core assembly obtained by cryo-EM cannot provide an ultimate answer to protein-protein interactions in 2-oxo acid dehydrogenase complexes. To address these issues, integrative structural approaches will be required.

## 7. Conclusions and Perspectives

The 2-oxoacid dehydrogenase complexes are macromolecular assemblies functioning in different metabolic pathways. These assemblies provide high catalytic efficiency and high-level of regulation to the metabolic pathways in the cell and the regulation of gene expression in the nucleus. Recent experimental findings in vitro and in vivo have identified the formation of hybrid complexes between the OGDHc and E1a, suggesting functional crosstalk between the two distinct metabolic pathways and an additional layer of regulation that needs to be thoroughly established. The direct association of the OGDHc and/or OADHc with known histone acetyltransferase KAT2 and histone H3/H4 has recently been discovered, suggesting a functional crosstalk between the KAT2 and OGDHc/OADHc in nuclei for the purpose of posttranslational modification of histones by succinylation/glutarylation and regulation of gene expression, tumor cell proliferation, and tumor formation [31,32]. However, direct involvement of the OGDH and OADHc in succinylation/glutarylation of mitochondrial and nuclear proteins needs to be explored. While over the past several years the importance of the OGDHc and OADHc in human health and disease, and their role as potential therapeutic targets have been increasingly emphasized, recent studies defining their structural organization and assembly into macromolecular machines are limited by cryo-EM reconstruction of the E2o inner core. In the absence of atomic resolution structure of any of the 2-oxo acid dehydrogenase complexes, recent advances in chemical cross-linking coupled with mass spectrometry, and in computational data processing have emerged as powerful tools to elucidate the potential interactions in binary E1o-E2o and E1a-E2o assemblies, and to identify their binding surfaces as well as their relative orientation via molecular modeling. Recent advances in negative stain EM and in single particle cryo-EM are promising to determine the pseudo-atomic structural models for assembly in 2-oxoacid dehydrogenase complexes. The combination of biochemical, biophysical and structural approaches by using cross-linking MS and cryo-EM structural biology is particularly promising to provide vital information for the assembly of 2-oxoacid dehydrogenase complexes, their function and regulation. At the same time, such studies will create a scaffold onto which novel inhibitors could be designed to mitigate the consequences of diseases associated with dysfunction of the TCA cycle or lysine degradation.

## Data Availability

The mass spectrometry proteomics data have been deposited to the ProteomeXchange Consortium (www.proteomexchange.org) via the PRIDE partner repository with the dataset identifier PXD017792 (accessed on 21 April 2020) and PXD023525 (not announced yet).

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
