# Peer review of "Toward an Understanding of the Structural and Mechanistic Aspects of Protein-Protein Interactions in 2-Oxoacid Dehydrogenase Complexes"

_life, 2021, doi:10.3390/life11050407_

Round 1

Reviewer 1 Report

The manuscript “Toward an understanding of the structural and mechanistic aspects of protein-protein interactions in 2-oxo acid dehydrogenase complexes” submitted by Natalia S. Nemeria and coworkers dissects the molecular mechanism and protein-protein interactions in 2-oxo acid dehydrogenase complexes. In the manuscript the authors extensively reviewed the current state of the art of such complexes, based on recently published Cryo-EM and X-ray crystal structures. Overall, the manuscript is well presented, organized and the topic reviewed is interesting for the field.

Minor comments and suggestions:

1)     Abstract. Please define TCA in the abstract.

2)   Scheme 1. Please improve the quality of the scheme and its overall resolution. The current presentation of this scheme is poor.

3) I would suggest to the authors to revise the resolution in the other Figures and improve if necessary. For instance, Figure 3, panels A and B have a low ressolution.  Overall, the size of the labels should be increased to facilitate their visibility to the readers.

Author Response

Response to Reviewer 1 Remarks

Reviewer 1

The manuscript “Toward an understanding of the structural and mechanistic aspects of protein-protein interactions in 2-oxo acid dehydrogenase complexes” submitted by Natalia S. Nemeria and coworkers dissects the molecular mechanism and protein-protein interactions in 2-oxo acid dehydrogenase complexes. In the manuscript the authors extensively reviewed the current state of the art of such complexes, based on recently published Cryo-EM and X-ray crystal structures. Overall, the manuscript is well presented, organized and the topic reviewed is interesting for the field.

Minor comments and suggestions:

1)  Abstract. Please define TCA in the abstract.

     Response: the TCA in the Abstract is defined now as ‘tricarboxylic acid cycle’

2)  Scheme 1. Please improve the quality of the scheme and its overall resolution. The current

     presentation of this scheme is poor.

     Response: The Scheme 1 was redone to improve its quality and resolution as

     recommended.

3)  I would suggest to the authors to revise the resolution in the other Figures and improve if

     necessary. For instance, Figure 3, panels A and B have a low resolution.  Overall, the size

     of the labels should be increased to facilitate their visibility to the readers.

Response: The resolution of Figure 3, panels A and B has been improved by inserting

new panels and by enlarging them. To accomplish this, the domain structure of the E1a subunit on the top of Figure 3 was removed as it is not discussed in the text.  Also, it needs to be mentioned that panels A and B are reprinted from the publication as mentioned under the Fig. 3.

legend with the quality available from the reprinted journal.

Reviewer 2 Report

The review presented by Nemeria et al is well written and presents an in-depth analysis of the 2-oxo acid dehydrogenase complexes. In general I have no reservations in accepting the paper as it is, it may however benefit from a few minor comments that I have.

  • The paper takes on the gigantic task of describing a huge chunk of literature in a review, while I certainly appreciate this, in my opinion the manuscript has become way too complex for people outside the field to comprehend. Any attempts of lowering the technical jargon, making the manuscript more comprehensible to general audience would be highly appreciated.
  • Page 4, Line 8 and Line 35. If the authors agree they might consider making two sub headings within the section 2 of the manuscript describing the proposed mechanisms for reductive acylation and succinyl-CoA synthesis.
  • Similarly in Page 7, 2nd para could have its own sub heading.
  • Figure 2B, 3B the images are not at all clear, I suggest the authors retrace/redraw the protein/surrounding density map using a standard molecular viewer like pymol/chimera etc.
  • I appreciate the efforts taken in the modelling described in Figure 6. I think it provides very relevant models which might be of extreme benefit to the people researching in this area. I think the manuscript will benefit from having a larger figure 6B and 6c. Also the authors might want to present cutouts of the complex in an open book format highlighting the interaction site as mapped onto E2o and E1a.
  • Figure 7B. Green and Yellow are not contrasting well, similarly orange and salmon are difficult to distinguish. The authors might consider generating a new image for 7b. Additionally if the authors find it useful they might want to come up with a rough/crude cartoon depiction of this interaction.

Author Response

Reviewer 2 Remarks

The review presented by Nemeria et al is well written and presents an in-depth analysis of the 2-oxo acid dehydrogenase complexes. In general, I have no reservations in accepting the paper as it is, it may however benefit from a few minor comments that I have.

  • The paper takes on the gigantic task of describing a huge chunk of literature in a review, while I certainly appreciate this, in my opinion the manuscript has become way too complex for people outside the field to comprehend. Any attempts of lowering the technical jargon, making the manuscript more comprehensible to general audience would be highly appreciated.

            Response: We gratefully accepted the Reviewer’s recommendation to introduce more  sub-sections as noted below. Also, all Figures and Scheme 1 mentioned by the Reviewers have been redone. Both of these actions on our part will make the review easier to appreciate. The review has been proof-read multiple times, also rewritten in large parts.

             As to the ‘jargon’, unfortunately we cannot alter what was suggested in the original papers. We do present an extensive list of abbreviations.

  • Page 4, Line 8 and Line 35. If the authors agree they might consider making two subheadings within the section 2 of the manuscript describing the proposed mechanisms for reductive acylation and succinyl-CoA synthesis.

Response: We appreciate the Reviewer’s remark as it makes the review more organized. In accord with the Reviewer’s remark, two subheadings were inserted on page 4:  subheading 2.1.1. ‘Mechanism for the reductive acylation reaction’ and the subheading 2.1.2. ‘Synthesis of succinyl-CoA’.

  • Similarly in Page 7, 2ndparagraph could have its own subheading.

Response: Similarly, two subheadings were inserted on page 7: 2.2.1.’Interactions in binary E1o-E2o, E1o-E3 and E2o-E3 subcomplexes’ and 2.2.2. ‘The assembly of E3 into OGDHc’.

  • Figure 2B, 3B the images are not at all clear, I suggest the authors retrace/redraw the protein/surrounding density map using a standard molecular viewer like pymol/chimera etc.

Response: Figure 2B and Figure 3B were redone to improve images. Fig. 2B and Fig. 3B are reproduced from publications with the resolution available in those publications.

  • I appreciate the efforts taken in the modelling described in Figure 6. I think it provides very relevant models which might be of extreme benefit to the people researching in this area. I think the manuscript will benefit from having a larger figure 6B and 6c. Also, the authors might want to present cutouts of the complex in an open book format highlighting the interaction site as mapped onto E2o and E1a.

Response: Figures 6B and 6C were enlarged in accord with the Reviewer’s remark.

  • Figure 7B. Green and Yellow are not contrasting well, similarly orange and salmon are difficult to distinguish. The authors might consider generating a new image for 7b. Additionally, if the authors find it useful, they might want to come up with a rough/crude cartoon depiction of this interaction.

Response:  The color in Figure 7B was modified in accord with the recommendation. The cartoon will be depicted in future studies once more information is available for the E1a-E2o and E1o-E2o assembly.